# Mosaicking Weather Radar Retrievals from an Operational Heterogeneous Network at C and X Band for Precipitation Monitoring in Italian Central Apennines

**Stefano Barbieri** [1,2,*], **Saverio Di Fabio** [1], **Raffaele Lidori** [1], **Francesco L. Rossi** [3], **Frank S. Marzano** [1,2] and **Errico Picciotti** [1]

1  CETEMPS, Department of Physical and Chemical Sciences, University of L'Aquila, 67100 L'Aquila, Italy; saverio.difabio@aquila.infn.it (S.D.F.); raffaele.lidori@himet.it (R.L.); frank.marzano@uniroma1.it (F.S.M.); errico.picciotti@aquila.infn.it (E.P.)
2  DIET, Department of Information Engineering, Electronics and Telecommunications, Sapienza University of Rome, 00184 Rome, Italy
3  CFA, Civil Protection Functional Centre of Abruzzo Region, 67100 L'Aquila, Italy; francesco.rossi@regione.abruzzo.it
*  Correspondence: stefano.barbieri@uniroma1.it; Tel.: +39-06-44585847

**Abstract:** Meteorological radar networks are suited to remotely provide atmospheric precipitation retrieval over a wide geographic area for severe weather monitoring and near-real-time nowcasting. However, blockage due to buildings, hills, and mountains can hamper the potential of an operational weather radar system. The Abruzzo region in central Italy's Apennines, whose hydro-geological risks are further enhanced by its complex orography, is monitored by a heterogeneous system of three microwave radars at the C and X bands with different features. This work shows a systematic intercomparison of operational radar mosaicking methods, based on bi-dimensional rainfall products and dealing with both C and X bands as well as single- and dual-polarization systems. The considered mosaicking methods can take into account spatial radar-gauge adjustment as well as different spatial combination approaches. A data set of 16 precipitation events during the years 2018–2020 in the central Apennines is collected (with a total number of 32,750 samples) to show the potentials and limitations of the considered operational mosaicking approaches, using a geospatially-interpolated dense network of regional rain gauges as a benchmark. Results show that the radar-network pattern mosaicking, based on the anisotropic radar-gauge adjustment and spatial averaging of composite data, is better than the conventional maximum-value merging approach. The overall analysis confirms that heterogeneous weather radar mosaicking can overcome the issues of single-frequency fixed radars in mountainous areas, guaranteeing a better spatial coverage and a more uniform rainfall estimation accuracy over the area of interest.

**Keywords:** weather radar; networking; mosaicking algorithm; data processing; validation

## 1. Introduction

In mountainous regions, heavy rainfall represents a problem that can manifest itself in the form of flash floods, especially in relatively small river basins. The Abruzzo region in central Italy is characterized by complex orography with vast mountainous regions within the Apennine range (with its highest peak, the Gran Sasso, at 2912 m above the sea level) and several catchments with very rapid runoff [1]. The detection and warning of severe events are typically approached using both rain gauges (RGs) and remote sensing instruments, from the ground and from space, in order to obtain a quantitative precipitation estimation (QPE) as accurately as possible [2–8]. In this respect, the ground-based microwave weather radar (WR) monitoring of precipitation systems is a well-established technique (e.g., [2,3]). It has a number of advantages over other instruments, as it can provide QPE products with

a high spatial resolution over a large area (i.e., hundreds of kilometers) within a relatively short period of time (i.e., a few minutes). Weather radar data are usually complemented by rain gauge networks, providing point-like surface rainfall measurement, as well as by satellite-based radiometers [4]. The latter, indeed, can provide cloud coverage and deep convection maps at infrared wavelengths but with a poor resolution, especially from geostationary platforms (i.e., many kilometers) and a poor temporal repetitiveness from low-Earth-orbit (LEO) platforms (i.e., few overpasses per day) [3].

Numerous sources of errors may affect single radar measurements, such as ground clutter backscattering, incomplete vertical profiling, non-uniform beam filling, two-way path attenuation, mixed-phase hydrometeors, anomalous propagation, and second-trip echos (e.g., [2]). The use of radar in mountainous terrain is particularly affected by the partial blocking of the radar antenna beam and the range-dependent degradation of surface rainfall retrieval due to increasing altitude and beam broadening. Operationally speaking, to improve the accuracy of radar QPE while preserving their spatial description of rainfall fields, many approaches suggest adjusting radar QPEs as a function of rain gauge measurements [4]. In this way it is possible to combine the advantages of in-situ sensors with radar remote sensing ones, partially overcoming their respective drawbacks [5]. Several radar–rain-gauge merging techniques have been developed, which have proven effective to improve the accuracy of single-radar QPEs (e.g., [4,6,7]) such as geostatistical approaches, based on the Kriging spatial extrapolation method, and simpler algorithms, based on an anisotropic correction map (e.g., [8]).

When a radar network is available within the region of interest, the multi-radar data composite techniques allow obtaining better quantitative rainfall retrieval than those obtained with individual radars [9–17]. This is due not only to the better spatial coverage, especially near the surface, but also to the improved quality of data products in overlapping areas. Several meteorological services and research centers across the world are using networks of weather radars both at the national and regional scales, e.g., [9–12]. These networks are very often heterogeneous so that different frequencies, antenna geometries, and scanning strategies pose some challenges when trying to mosaic their data [13,14]. Although the use of X-band weather radars has been successfully demonstrated for many applications, their use in coordination with C-band radars still needs a detailed and systematic analysis for an operational radar network, especially when dual polarization is not available and they are operated in complex-orography regions [15–17]. The radar composite products can be generated combined onto a unified 3D Cartesian grid; the volume of data generated from each radar, from an operational point of view, is more effective and straightforward [18] to use in generating products at the single-radar level and then combine them onto a unified 2D Cartesian grid.

The Abruzzo weather radar network design was driven by the needs of the Functional Centre of Region Abruzzo (CFA) for the detection and warning of severe weather and related hydrometeorological hazards, a requirement that requires high redundancy, availability, and accuracy of radar data. CFA is part of the Civil Protection Regional System of the Abruzzo Region in central Italy, which has functions regarding the forecast, monitoring, and warning of meteorological, hydrological, hydraulic, and wildfire risk, with the institutional duty of daily operational meteorological surveillance [16]. The Center of Excellence CETEMPS of the University of L'Aquila (Italy) carries out research activity, mainly concerning atmospheric physics, remote sensing, meteorology, and hydrology and supports CFA in its daily operational meteorological forecast and surveillance. The Abruzzo radar network, planned by CFA together with the Center of Excellence CETEMPS (L'Aquila, Italy) and completed in 2018, is based on X- and C-band systems, installed in different periods by diverse manufacturers, and performing different scanning strategies and signal processing [17,19].

Within the context just discussed, the purposes of this work are: (i) to introduce an operational three-dimensional (3D) radar raw-volume-processing chain, called RAMP (radar advanced multiband processing), to compensate for the most common error sources

for each system at the C and X bands with and without polarimetric capability and to provide single-radar-level products; (ii) describe the operationally-oriented radar composite modular algorithm, called CRAMS (CETEMPS radar advanced mosaic software), capable of ingesting both radar RAMP-processed products and providing 2D composite products; (iii) apply CRAMS to a large set of precipitation events in central Italy's Apennines and the Adriatic coast over 2 years and validate its surface rain-rate hourly rainfall output against the rain gauge data of the Abruzzo network by ranking the various 2D mosaicking schemes in terms of error index statistics.

This paper is organized as follows. In Section 2, the Abruzzo Region weather radar and rain gauge network is described in terms of C-band and X-band radars and available rain gauges. Section 3 presents the multi-radar mosaicking methodology wherein single-radar products for composite, radar–gauges adjustment methods, and mosaicking techniques are presented. The radar pattern mosaicking validation using rain-gauge data is illustrated in Section 4 with an overview of available case studies, the analysis of some selected events, and a discussion of the overall error statistical metrics for each considered technique. Conclusions are drawn in Section 5, whereas the Appendix A provides some details about the RAMP processing chain.

## 2. Abruzzo Region Weather Radar and Rain Gauge Network

The Abruzzo radar network consists of three weather radars: a C-band system located at the site of Mt. Midia (L'Aquila, Italy) and two X-band mini radars, both located along the Adriatic Sea coast, in Cepagatti (Pescara, Italy) and Tortoreto (Teramo, Italy), respectively, as shown in Figure 1.

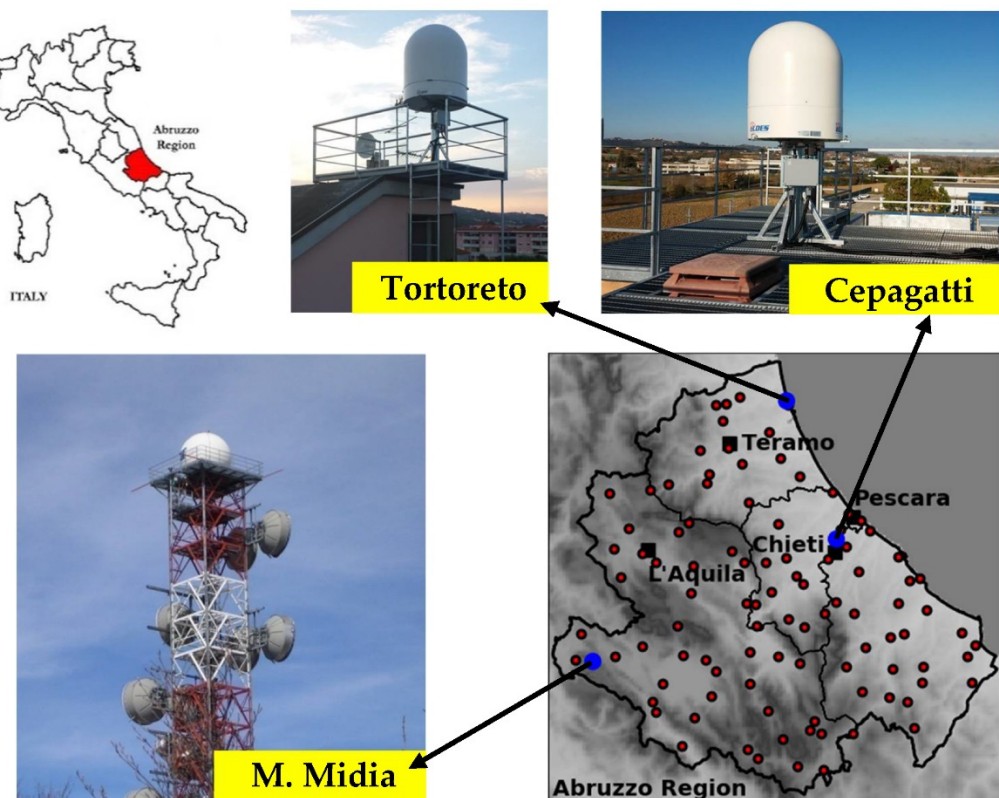

**Figure 1.** Locations and pictures of the weather radars at the C and X bands, installed within the Abruzzo region in central Italy (blue dots) and the location of the 98 quality-controlled rain gauges in the Abruzzo region (red dots).

The sites of all radar systems were chosen to guarantee the maximum extension of the composite taking into account technical, financial, and logistic constraints. The Abruzzo rain-gauge network, here considered to evaluate the QPE performance of the WR system, is composed of 98 quality-controlled stations (see Figure 1).

Both Abruzzo region weather radar and rain gauge features are described in the following two paragraphs.

### 2.1. C-band and X-band Weather Radars in Abruzzo

The current position of the Mt. Midia C-band Doppler weather radar dates back to 2006, when it was transferred from the previous site near the city of L'Aquila (see Figure 1). The radar project involved, in a synergistic work, the Italian Department of Civil Protection (DPC) as a sponsor, CETEMPS as coordinator, CFA, and some Italian companies in the sector [15].

The Mt. Midia radar is currently part of the Italian operational weather radar network [16], composed of 23 systems, managed by a federation of national and regional bodies. The Mt. Midia site is in a mountainous area at the border between the Abruzzo and Lazio regions. Mt. Midia's top height is 1710 m, thus covering most of central Italy, including the Abruzzo inland and the urban area of Rome, but leaving the Abruzzo coastline basically uncovered.

For this reason and with the aim of obtaining qualitatively valid observations along the Abruzzo coastline, two X-band radar systems were also installed in Abruzzo in the framework of three European projects, HYDRORAD [17], AdriaRadNet [19], and CapRad-Net [20]. The first one, with Doppler dual-polarized radar technology, became operational in 2014 at Tortoreto, whereas the other one, with non-Doppler single-polarization capability, was located in 2017 at Cepagatti, a few kilometers from the Pescara airport.

The technical specifications of all 3 weather radar systems installed in the Abruzzo region are shown in Table 1, whereas their locations and pictures are given, as mentioned, in Figure 1. Both the Tortoreto and Mt. Midia radars are typically operated with a range resolution of 125 m, while the Cepagatti radar uses a resolution of 450 m.

**Table 1.** Technical specifications of the three weather radars, installed in the Abruzzo region.

| Name/Features | M. Midia (MM) | Tortoreto (TO) | Cepagatti (CE) |
|---|---|---|---|
| Owner | CFA | CFA | CFA |
| System model | DWSR-93C | WR-25XP | WR-10X |
| Manufacturer | Enterprise, USA | ELDES, IT | ELDES, IT |
| Latitude | 42.06° | 42.78° | 42.40° |
| Longitude | 13.18° | 13.94° | 14.14° |
| Height (a.s.l.) | 1710 m | 15 m | 50 m |
| Polarization | Single | Dual | Single |
| Frequency band | C | X | X |
| Doppler capability | Yes | Yes | No |
| Peak power | 250 kW | 25 kW | 10 kW |
| Beamwidth | 1.6° | 3.0° | 3.0° |
| Antenna gain | 40.5 dB | 35 dB | 35 dB |

The volume scan period is the same for all radars and occurs every 10 min. Due to its altitude, Mt. Midia radar scans at 0.5°, 1.5°, 2.5°, and 3.5° elevation angles, whereas both the Tortoreto and Cepagatti radars are programmed to scan at 1°, 2°, 3°, 4°, 5°, and 30° and 1°, 2°, 3°, 4°, 5°, and 15°, respectively (the difference in the last elevation is due to the surrounding orography).

### 2.2. Rain Gauge Network in Abruzzo

The term rain gauge is generally used to refer to instruments used to measure the amount of liquid precipitation at a point location, over a set period of time, mainly by means of direct rainfall collection. Rain gauges are the most widely used rainfall sensors, providing low-cost, direct, and relatively accurate measurement. Rain gauge records are often used as ground truth in the calibration and verification of remote rainfall sensors such as weather radars [4–7]. Tipping bucket rain gauges (TB-RGs) are the most common type of automatic recording gauge and are in fact the main data source used for adjustment of radar rainfall estimates [6,21–23].

The Abruzzo region was chosen as a testing domain for both single-radar and composite precipitation estimates, whereas the rain-gauge network nominally consists of 98 TB-RGs stations, as shown in Figure 1. As is well-known, the bucket resolution of TB-RGs and the minimum hourly rainfall accumulation time period (e.g., 0.2 mm and 10 min for the Abruzzo network) can affect the effective accuracy and temporal resolution of the rain-gauge estimates [24]. The rain-gauge network data are collected and centrally processed in near-real time by DPC and distributed to functional centers and competence centers [25] via the DEWETRA data portal. The data are stored on DEWETRA in the form of hourly accumulations for further analysis.

The point-like rain-gauge records may contain significant instrumental errors arising from a variety of sources. In order of general importance, systematic errors common to all rain gauges, wetting loss in the internal walls of the collector, evaporation from the container, and errors due to in and out-splashing of water [26]. Additional errors in rain-gauge measurements may arise from operational conditions such as miscalibration, blockages, and double-tipping (in the case of TB-RGs). Operational errors can usually be detected (and therefore removed or corrected) through time-series analysis or through comparison against neighboring gauges and even against co-located radar estimates.

Given that rain-gauge records are often used as ground truth, all rain-gauge data employed in this work were preliminarily subjected to quality control in order to identify and remove any inconsistencies before being used in radar-rain gauge comparison. Typical quality checks are: (i) identification of rain gauges with the same name but different coordinates, (ii) removal of data associated with rain gauges without valid coordinates, iii) removal of duplicate data and identification of anomalous data (for example very different values compared to the surrounding rain gauges).

In this study, systematic, operational, and additional errors, highlighted during a quality check of rain-gauge data, were corrected by removing the measurement or the rain gauge itself from the comparison.

### 3. Single Radar and Mosaicking Methodology

The need to create a meteorological radar network arises both from the requirement for meteorological monitoring over a wider area and from the need to improve the quality of single-radar measurements and associated rainfall retrievals. As a matter of fact, the use of a single weather radar involves a series of problems that limit its effectiveness, such as areas within the nominal radar's unambiguous range not electromagnetically detectable due to the local orography or, especially at the X band, to strong precipitation leading to a considerable path attenuation or even the complete extinction of the backscattered signal. In the latter case, attenuation signal correction can be approached by using spatial coverage redundancy with additional radar sources not experiencing the same signal loss. To do this, we need to design a weather radar network and a scheme to generate a composite at the price to add new systems that can indeed extend the precipitation monitoring area itself.

The essential requirements for generating an accurate composite are: (i) control of the data quality of individual radars, using proper filtering and correction algorithms, (ii) mapping of the artifact-corrected radar data onto a uniform common grid with a homogeneous presentation, and the (iii) synchronization of the various radar sources with their temporal availability almost in real time. In summary, generating weather radar

composite products from multiple radar scans, having different system characteristics (e.g., see Table 1 in the case of the Abruzzo region), can offer a number of advantages when compared to observations of a single radar, summarized as follows:

- Coverage of a wider area of the monitored territory
    - overcoming the limited coverage of individual radars;
    - more accurate measurements at greater distances;

- Spatial redundancy of weather radar observations
    - inter-calibrating the hybrid network radars;
    - reducing ground clutter effects and beam blocking;
    - guaranteeing measurements even in the case of the failure of one radar;

- Mitigation of path attenuation effects and improved retrievals
    - reducing two-way path attenuation effects in complex orography;
    - extracting radar data in total signal extinction area;
    - improving the rainfall estimate thanks to simultaneous observations;

*3.1. Single-Radar Algorithms*

The realization of the multi-radar composite for the Abruzzo region involves both the use of complete volumetric scans and two-dimensional products from the Mt. Midia (MM), Cepagatti (CE), and Tortoreto (TO) radars, all synchronized and collected every 10 min. The 3D volume scans of each radar system are processed by means of the RAMP processing chain in the native radar (spherical) coordinates, briefly described in the Appendix A.

As mentioned, the results of each weather radar scan are often associated with significant uncertainties, arising from various sources: hardware calibration, ground clutter, range-related effects, beam blockage, beam overshooting or partial beam filling, anomalous propagation of the radar beam, and wireless local-area-network (WLAN) interferences. In practical applications, it is not possible to separate and estimate the different sources of radar errors, but they are still important to identify and characterize to improve the quality of the final products.

RAMP is a modular algorithmic platform able to harmonize and process data from radars with different specifications, deal with the most common error sources for each system, and generate products (see Appendix A). The RAMP schematic processing chain is shown in Figure 2 top panel, where the input and output data are highlighted. The input data consist of horizontally reflectivity $Z_{hh}$ and, for polarimetric radar, also differential reflectivity $Z_{dr}$ and differential phase shift $\phi_{dp}$. The output data consists of the same input variables, corrected by errors, and the products described in Table 2.

The RAMP approach is similar to that adopted in literature (e.g., [27,28]), in which the data processing is functionally divided into two steps: the first is aimed at raw data corrections, whereas the second is devoted to the characterization of radar data quality. The latter is expressed as a quantitative quality index, subjectively identified by the combination of several indicators and applied to each cell of the radar scan volume. Quality control algorithm functions can be switched on or off in the RAMP scheme.

Data-correction algorithms are essential for improving estimation uncertainty, whereas quality-control algorithms generate a total quality index (TQI) map that can be attached to radar-based retrievals [29]. Figure 2 (bottom panel) shows an example of the RAMP chain application to weather radar data in terms of data before and after RAMP corrections as well as attached TQI maps. The individual radars are not temporally synchronized so that a posterior synchronization is carried out both for volumetric 3D and 2D products.

A list of typical products, obtained from the RAMP algorithm for each individual radar, is given in Table 2 (see also Appendix A). All single-radar algorithms, organized in a series of sub-packages characterized by their specific functionality, have been developed to manage a variety of routines, including the reading, processing, analyzing, composing, and displaying of data from different weather radars with different input formats.

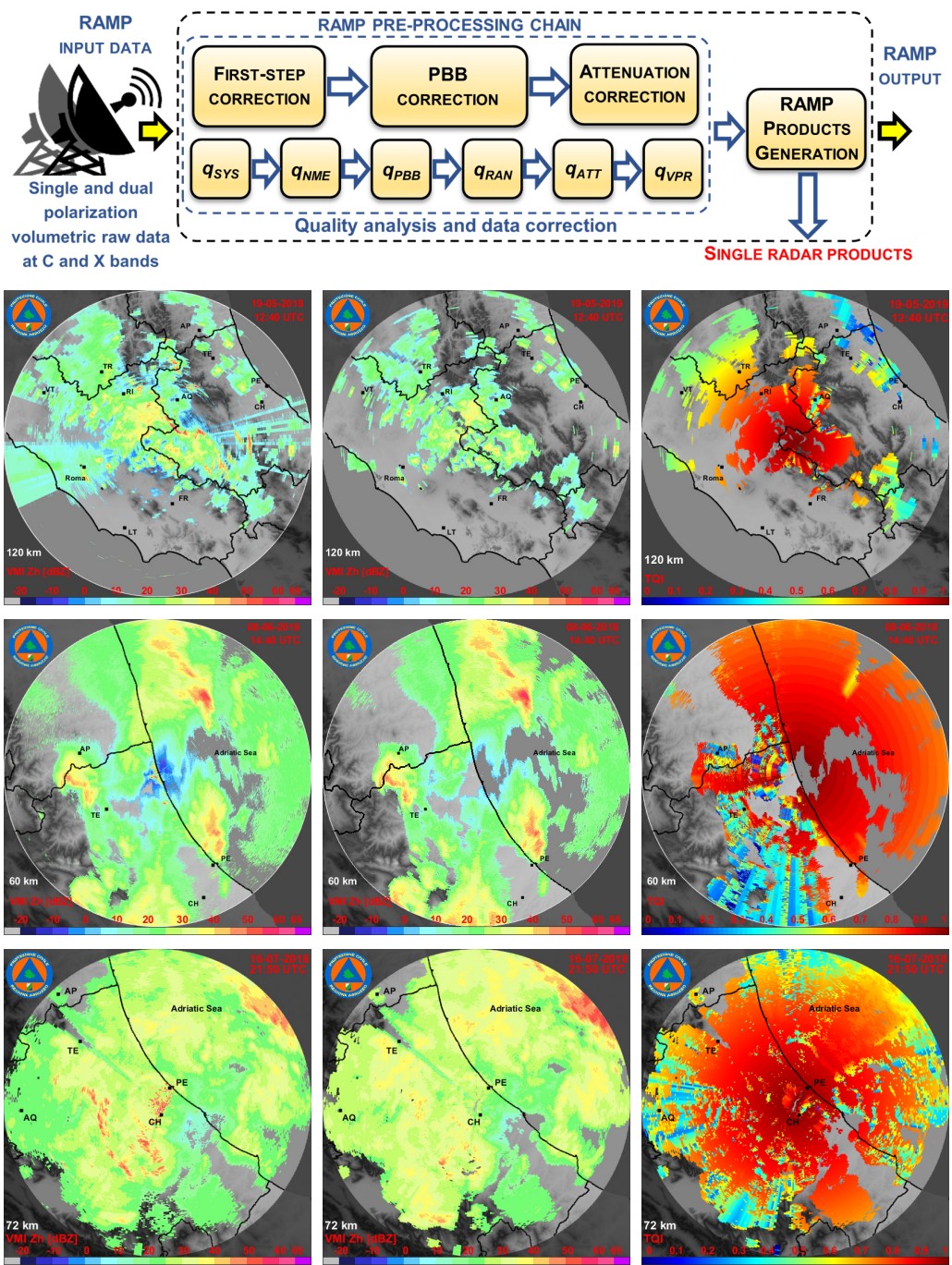

**Figure 2.** (**Top**) RAMP chain scheme and (**bottom**) an example of RAMP application to the Mt. Midia (MM), Tortoreto (TO) and Cepagatti (CE) radars: VMI product before (**left**) and after (**center**) data correction as well as the total quality index (TQI) is also shown (**right**).

In this work, as mentioned, we focus on the surface rain rate mosaicking. To this purpose, the horizontally reflectivity factor $Z_{hh}$ is usually converted into surface rainfall intensity (SRI) by means of a conventional power law of the form. In complex orography, we can approach the estimate using the lowest detectable bin (LDB) $Z_{hh\text{-}LDB}$ (i.e., the radar bin which is detected at the lowest altitude along the pixel column) or the vertical maximum intensity (VMI) $Z_{hh\text{-}VMI}$ (i.e., the radar bin with maximum reflectivity along the pixel column). For operational purposes, VMI provides a clear signature of the rainfall ingesting to some extent the vertical variability of the radar returns. For this reason, within the single-polarization RAMP scheme, for a given pixel (*x*,*y*) and at a given time *t*, we

have adopted the following parametric estimator of the radar-based SRI of the hourly rain rate $R_{WR}$:

$$R_{WR}(x, y, t) = f_S[Z_{hh-VMI}(x, y, t)] = a_R [Z_{hh-VMI}(x, y, t)]^{b_R} \tag{1}$$

where $f_S$ is the single-polarization estimation function; $(x, y)$ are local grid Cartesian coordinates; $a_R$ and $b_R$ are empirical regression parameters derived from an external calibration using rain-gauge quality-controlled data in order to tackle the DSD-dependence in single-polarization systems (e.g., [30–32]). These coefficients may depend on the precipitation type (e.g., stratiform, convective, embedded) and season, but, for operational purposes, this data stratification is not applied due to the errors in classifying each radar bin within an event. In (1), the subscript WR stands for the MI, CE, or TO weather radar (see Table 1).

**Table 2.** List of the main products of each single radar using RAMP within the Abruzzo network composite. RAMP modules are briefly described in Appendix A.

| Description | Symbol |
| --- | --- |
| Vertical maximum of reflectivity. This product is useful for the quick surveillance of regions covered by the radar. | VMI |
| Convective storm detection. This product is aimed at distinguishing stratiform and convective precipitation. | CSD |
| Nowcasting. This product is aimed at a short-term forecast of convective cells' motion. | NOW |
| These products estimate the ground instantaneous (SRI) and accumulated (SRT) rain over the radar coverage area. | SRI, SRT |
| Vertically integrated liquid. This product can be used as a measure of the potential for strong rainfall. | VIL |
| Probability of hail. This product is aimed at the detection of hail, which is one of the most dangerous weather phenomena. | POH |

In the case of polarimetric radars, such as the TO radar at the X band in the Abruzzo network (see Figure 1), additional variables are available (e.g., differential reflectivity $Z_{dr}$ and differential phase shift $\Phi_{dp}$), so that horizontal specific attenuation $A_{hh}$ can be derived to correct for two-way path extinction effects. In this case, we expect an improved rainfall rate estimation, providing estimates of DSD fitting parameters [30]. Alternatively, robust estimators of SRI, based on the specific differential phase shift $K_{dp}$ and the differential reflectivity $Z_{dr}$ can be applied, especially in the case of moderate to intense precipitation after correcting for two-way path attenuation (see the Appendix A):

$$R_{WR}(x, y, t) = f_P[Z_{hh-VMI}(x, y, z, t), K_{dp}(x, y, z, t), Z_{dr}(x, y, z, t)] \tag{2}$$

where $f_P$ is the dual-polarization estimation function involving the vertical maximum intensity of the corrected reflectivities $Z_{hh-VMI}$ and $Z_{dr}$, as well as the estimated $K_{dp}$ [27]. Specific modules are introduced in the RAMP scheme to process polarimetric radar observables and to generate further products such as the hydrometeor classification for each radar volume scan (see Appendix A).

As mentioned, the main steps of data correction and quality characterization within the RAMP chain are briefly summarized in the Appendix A. To mitigate the range-dependent degradation of radar rain estimates, for operational purposes we have adopted a static radar-gauges adjustment methodology which will be described in some detail in the next section.

### 3.2. Radar-Gauge Spatial-Adjustment Methods

For operational purposes, radar-gauge adjustment methods attempt to correct the average error, which is always present in radar-based hourly rain rate $R_{WR}$ when compared to rain gauge hourly rain rate $R_{RG}$ within the single-radar coverage. The main statistical

approach in literature uses an isotropic multiplicative factor, and they are well resumed in [33]; however, an anisotropic map approach [8] has also been used here.

In the first method, the whole radar $R_{WR}(r)$ field is used as a background, applying a multiplicative (or additive in some cases) correction factor. The isotropic-range adjustment (IRBA) algorithm defines the radar adjustment factor as a space–time average ratio of rain gauge and co-located radar rain rates over the considered time period [8]. Resampling the radar estimates at a 0.400-km resolution in a Cartesian grid $(x,y)$ and representing the IRBA algorithm as a function of the radar range $r(x,y)$, the average radar adjustment factor $F_{adj}(x,y)$ can be expressed as an exponential function of the range:

$$F_{adj}(x,y) = F_{IRBA}(r) = \langle \frac{R_{RG}(r)}{R_{WR}(r)} \rangle = c \, e^{\, d \, r} \tag{3}$$

where the angle brackets indicate the space–time average, whereas $c$ and $d$ are empirical regression coefficients, derived from the comparison between rain gauge and weather radar usually at the available rain-gauge locations over a time period of two years (2018–2019).

For the Mt. Midia radar, an example of IRBA output map is presented in Figure 3a. The IRBA factor is basically a radial scaling of the original radar-retrieved field using available rain-gauge stations not only in Abruzzo but also for central Italy regions covered by the Mt. Midia radar range of 120 km.

In the second method, unlike the isotropic adjustment approach, the anisotropic spatial adjustment (ASBA) method is assuming a spatially inhomogeneous correction factor, which means a range dependence that may vary not only with respect to radar range $r(x,y)$, but also to the azimuth $\varphi$. Since the radar grid is resampled at 1-km resolution, the rain gauge field is geospatially interpolated by means of the Kriging method. The Kriging interpolation consists of predicting values at ungauged locations as the linear combination of values at gauged locations, with the linear weights being derived through the minimization of the variance of the estimation error [34]. As such, the so-called Kriging yields the best linear unbiased estimator of the rainfall field, based upon available rain-gauge records. Indicating with $R_{RG\text{-}K}$ the Kriging-interpolated rain field, the ASBA factor can be formally expressed by:

$$F_{adj}(x,y) = F_{ASBA}(r, \varphi) = \langle \frac{R_{RG-K}(r, \varphi)}{R_{WR}(r, \varphi)} \rangle \tag{4}$$

The output map of the ASBA method, obtained using two years of rain-gauge data, is presented, for the Mt. Midia radar, in Figure 3b, wherein we note the azimuthal inhomogeneities between the ASBA map and the corresponding IRBA one. The ASBA approach leads to the generation of a correction adjustment map for each individual radar, which is then applied to the rainfall estimates before performing the radar product mosaicking.

In summary, the estimated rain rate fields $R_{WRadj}$ from each single weather radar can be expressed by applying the adjustment factor $F_{adj}$ (in one of its 2two forms given in (3) or (4)), as follows:

$$R_{WRadj}(x,y,t) = F_{adj}(x,y) \, R_{WR}(x,y,t) \tag{5}$$

where the considered weather radars are MM, TO, or CE (see Table 1), and the adjustment method can be IRBA or ASBA. The previous expression justifies the choice to perform the radar network mosaic in the rain-rate domain where we can uniformly merge data coming from systems at different frequencies (e.g., C or X bands) and having single- or dual-polarization, as in the case of the Abruzzo heterogeneous radar network. If no adjustment factor is adopted, then it remains $F_{adj}(x,y) = 1$.

### 3.3. Radar Mosaicking Techniques

This section describes the mosaicking algorithm CRAMS for the multi-radar heterogeneous network, developed for the generation of the regional-scale composite products [35]. The CRAMS technique, whose general scheme is shown in Figure 4, is designed to provide the necessary tools to build a complete radar processing chain in a flexible and modular

way to adapt to different radar networks. The single-radar RAMP products are remapped into a 3D Cartesian grid preserving height information, spatial resolution, and coverage range in order to initialize the CRAMS algorithm for the composite generation.

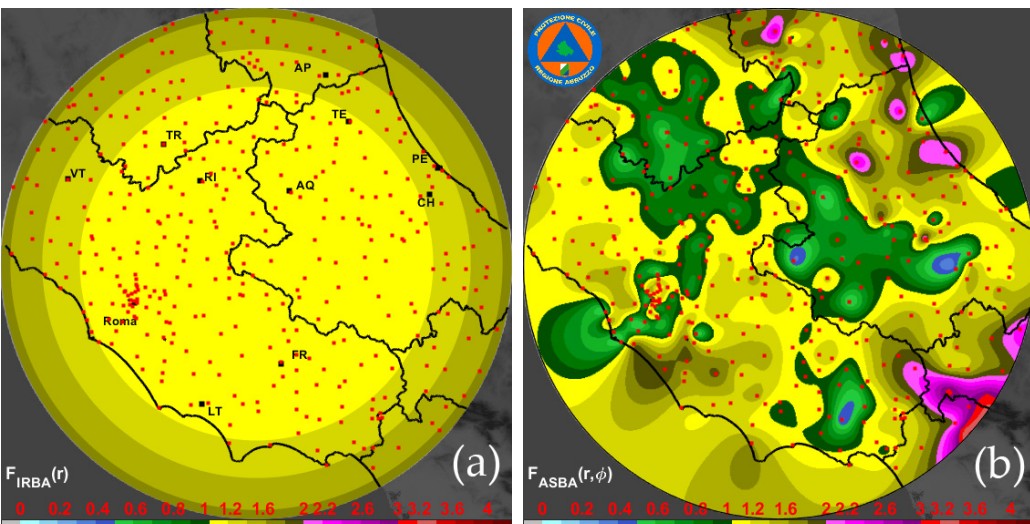

**Figure 3.** Geospatial correction map applied to the Mt. Midia weather radar, derived from two years of data (2018–2019). (**a**) Isotropic range adjustment (IRBA) method, (**b**) anisotropic spatial adjustment (ASBA) method. Red dots indicate the available rain gauge stations not only in Abruzzo but also for central Italy regions covered by the Mt. Midia radar range of 120 km.

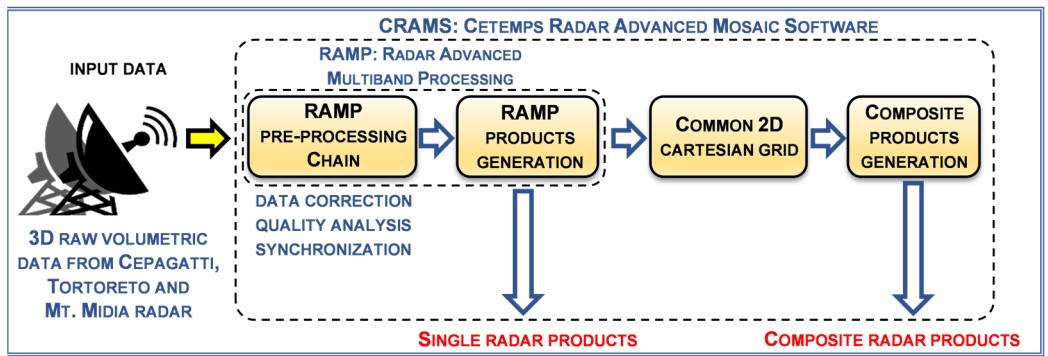

**Figure 4.** Overview of the CRAMS chain flowchart: all data are collected every 10 min and synchronized, and the related composite products are generated in real time as output maps.

The Abruzzo mosaicking domain of about 83,000 km$^2$ (310 km by 268 km) is set up considering the technical characteristics of each weather radar, as well as the geographical location and the surrounding orography. For each radar, the coverage area is set to be compatible with the size of the mosaicking domain, as shown in Figure 5a, covering a large part of central Italian territory. The WR products, generated at the single-radar level, are remapped onto a common Cartesian 2D grid with a spatial resolution of 0.400 km, covering the mosaicking domain, a trade-off able to guarantee sufficient accuracy, ease of implementation, and calculation speed in processing.

To represent the orography complexity and the impact of radar beam blockage, Figure 5 (right panel) also shows the map of the minimum LDB (lowest detectable bin) height, mosaicked from the corresponding maps of the three weather radars using the SRTM (shuttle radar topographic map) digital elevation model (DEM) of the Abruzzo region at 30-m resolution. The discontinuities at the border of WR maximum-range circles are mainly due to the orographic blockages dependent on the WR installation sites (along the coast for

the TO and CE radars and in the middle of the Apennines for the MM radar), as reflected by their altitudes in Table 1.

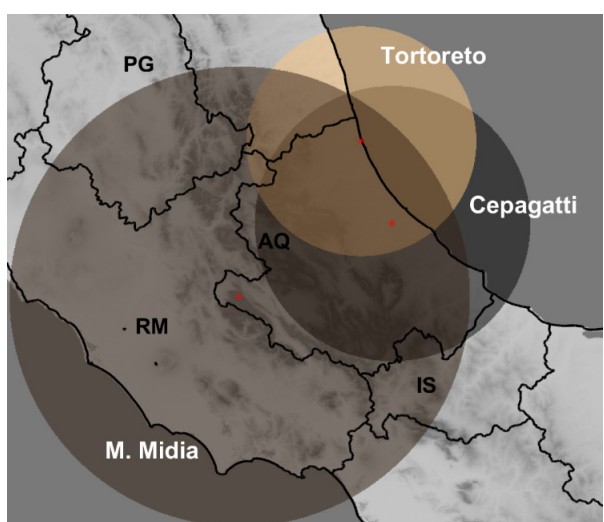 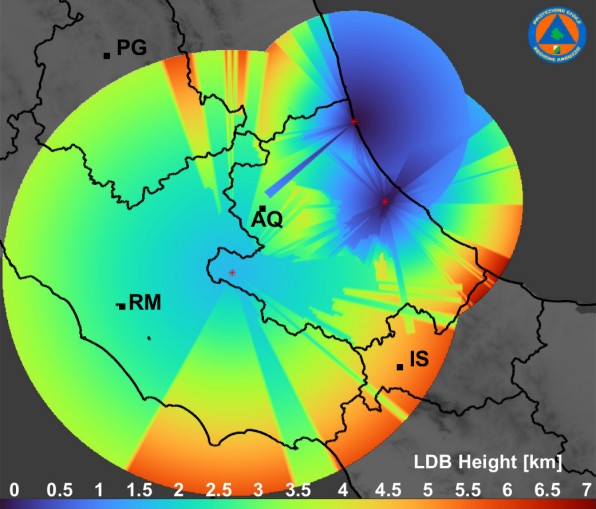

**Figure 5.** (**Left panel**) The Abruzzo network composite domain with the maximum coverage area chosen for each radar system (120 km for C-band MM, 80 km for X-band CE, and 60 km for X-band TO). (**Right panel**) Map of the minimum LDB (lowest detectable bin) height, mosaicked from the corresponding maps of the three weather radars represented by the operational maximum range (see the **left panel**).

The resampled radar products can be combined in order to obtain the final regional WR composite. Selecting an optimal way to merge radar data from different systems is a critical task and several factors, such as the size and shape of the domain of interest, the number of radars, and their geographic location should be considered. The WR composite can be generated directly from scanned volume data from each radar or from some specific WR-based products, generated by every single radar. In the first case, 3D observations from individual radars can be combined onto a unified 3D Cartesian grid, whereas in the latter case 2D products from individual radars can be combined onto a unified 2D Cartesian grid. The main differences between 3D and 2D mosaics are well described in [36].

The 2D composite scheme is adopted in the CRAMS algorithm. The 2D approach can be at a first glance less accurate than the 3D one, but it has the advantage of being computationally less complex and thus easier to implement in an operational environment, especially if the domain size is wide and the data to work with are heterogeneous since they come from radars with very different characteristics.

Formally speaking, the mosaicking technique is providing the weather radar network (WRN) rain rate estimation $R_{WRN}$ at a given time by means of:

$$R_{WRNet}(x,y,t) = M_{net}\Big\{ R_{MMadj}(x,y,t), R_{TOadj}(x,y,t), R_{CEadj}(x,y,t)\Big\} \quad (6)$$

where $M_{net}$ is the mosaicking technique rule for the network radar data, whereas $R_{MMadj}$, $R_{TOadj}$, and $R_{CEadj}$ are the estimated rainfall rate adjusted by applying the $F_{adj}$ factor to the MM, TO, and CE radars, respectively.

In addition to the rain estimation, the CRAMS mosaicking technique can also be applied to the other products of the individual radars, as shown later in Section 4.1 with an example. The discussion of these other mosaic products is outside the scope of this work.

After a literature review about 2D mosaicking, Table 3 summarizes the main rules $M_{net}$ for radar mosaicking. We basically considered 4 approaches in terms of:

1.  *Max*, wherein a multi-radar maximum criterion is used;
2.  *Avg*, wherein a multi-radar average criterion is used;

3.  *Lin*, wherein a multi-radar linear distance-weighted criterion is used;
4.  *Exp*, wherein a multi-radar exponential distance-weighted criterion is used.

**Table 3.** Merging strategies for multi-radar mosaicking, tested in the Abruzzo composite domain (see Figures 1 and 5). The identification (ID) number lists the merging (from 1 to 4) and processing (a or b) methods and then their combination as a merging technique (from 1a to 4ab). SRT is the surface rain total (mm), accumulated in time on a given site.

| ID | Label | Multi-Radar Merging Method | Reference |
|---|---|---|---|
| 1 | Max | Assign to the common pixel the maximum value of the available measurements. | [37] |
| 2 | Avg | Assign to the common pixel the mean value of the available measurements. | [37] |
| 3 | Lin | Assign to the common pixel a value-weighted with the distance from the radars, using linear weighting functions. | [38] |
| 4 | Exp | Assign to the common pixel a value weighted with the distance from the radars, using exponential weighting functions. | [37] |
| **ID** | **Label** | **Single-Radar Processing Method** | **ID LABEL** |
| a | Pol | Polarimetric rain rate estimation applied to the polarimetric radar in Tortoreto (TO). | P |
| b | Ani | Single-radar spatial anisotropic correction, based on the ASBA mapping (see Figure 4). | A |
| **ID** | **Label** | **WR Network (WRN) Mosaicking Technique** | **ID LABEL** |
| 1 | Max | Assign the maximum value among those covering the same grid of cells. | $R_{WRN1}$ |
| 1a | MaxPol | Assign the maximum value among the available measurements. Polarimetric processing is performed on the TO-radar. | $R_{WRN1a}$ |
| 1b | MaxAni | Assign the maximum value among the available measurements after a single-radar spatial anisotropic correction. | $R_{WRN1b}$ |
| 1ab | MaxPolAni | Assign the maximum value among the available measurements. A single-radar spatial anisotropic correction and polarimetric processing on the TO-radar is performed. | $R_{WRN1ab}$ |
| 2 | Avg | Assign the mean value among those covering the same grid of cells. | $R_{WR2}$ |
| 2a | AvgPol | Assign the mean value among the available measurements. Polarimetric processing on the TO-radar is performed. | $R_{WRN2a}$ |
| 2b | AvgAni | Assign the mean value among the available measurements after a single-radar spatial anisotropic correction. | $R_{WRN2b}$ |
| 2ab | AvgPol Ani | Assign the mean value among the available measurements. A single-radar spatial anisotropic correction and polarimetric processing on the TO-radar is performed. | $R_{WRN2ab}$ |
| 3 | Lin | Assign the value linear weighted with the distance from the radars among those covering the same grid of cells. | $R_{WRN3}$ |
| 3a | LinPol | Assign the value linear weighted with the distance among the available measurements. Polarimetric processing is performed on the TO-radar. | $R_{WRN3a}$ |
| 3b | LinAni | Assign the value linear weighted with the distance among the available measurements after a single-radar spatial anisotropic correction. | $R_{WRN3b}$ |
| 3ab | LinPol Ani | Assign the value linear weighted with the distance among the available measurements. A single-radar spatial anisotropic correction and polarimetric processing on the TO-radar is performed. | $R_{WRN3ab}$ |
| 4 | Exp | Assign the value exponential weighted with the distance from the radars among those covering the same grid of cells. | $R_{WRN4}$ |
| 4a | ExpPol | Assign the value exponential weighted with the distance among the available measurements. Polarimetric processing is performed on the TO-radar. | $R_{WRN4a}$ |
| 4b | ExpAni | Assign the value exponential weighted with the distance among the available measurements after a single-radar spatial anisotropic correction. | $R_{WRN4b}$ |
| 4ab | ExpPol Ani | Assign the value exponential weighted with the distance among the available measurements. A single-radar spatial anisotropic correction and polarimetric processing on the TO-radar is performed. | $R_{WRN1b}$ |

Moreover, any option can be switched on and off, such as the polarimetric retrieval for the TO radar as well as the anisotropic correction for all radars, thus setting up $F_{adj}(x,y) = 1$. The 16 combinations of all possible approaches, considered in this work, are labeled with numbers (1, 2, 3, and 4) and labels (a, b) to be more effective when referring to and comparing them during the validation stage of this work.

### 4. Mosaicking Validation Using Rain Gauge Data

The mosaicking techniques, listed in Table 3, have been tested on a series of case studies in the period 2018–2020, related to different precipitation regimes in the Abruzzo region, listed in Table 4.

**Table 4.** Precipitation test cases in 2018–2020 within the Abruzzo domain used for intercomparing and validating the merging techniques listed in Table 3. Duration and maximum rain rate are derived from the rain gauge time series over the whole domain in Figure 2.

| Date | Atmospheric Phenomena | Precipitation Type | Duration (Day) | Maximum Rain Rate (mm/h) |
|---|---|---|---|---|
| 3 May 2018 | Rainstorm | Moderate/Frequent | 1 | 49 |
| 8 May 2018 | Rainstorm | Moderate/Frequent | 1 | 58 |
| 5 June 2018 | Storm | Light/Discontinuous | 1 | 41 |
| 8 June 2018 | Rainstorm | Moderate-heavy/Frequent | 1 | 61 |
| 22 June 2018 | Storm | Moderate/Frequent | 1 | 60 |
| 6 July 2018 | Rainstorm | Light/Discontinuous | 1 | 56 |
| 16 July 2018 | Rainstorm | Moderate/Frequent | 1 | 35 |
| 14 August 2018 | Rainstorm with hail | Intense/Persistent | 1 | 83 |
| 5 May 2019 | Rainstorm with snow/hail | Light/Discontinuous | 3 | 44 |
| 10 July 2019 | Rainstorm with hail | Intense/Persistent | 1 | 72 |
| 4 February 2020 | Rain and snow | Moderate/Frequent | 2 | 37 |
| 27 March 2020 | Rain and snow | Light/Discontinuous | 1 | 14 |
| 3 May 2020 | Rainstorm | Light/Discontinuous | 1 | 38 |
| 17 July 2020 | Rainstorm with hail | Light/Discontinuous | 1 | 42 |
| 7 October 2020 | Rainstorm with hail | Moderate/Discontinuous | 1 | 75 |
| 20 November 2020 | Rain and snow | Moderate/Frequent | 1 | 52 |

The 16 selected rainfall cases include rainstorms and stratiform rain, as well as snowfall and hail occurrences during the rainiest seasons in central Italy, that is from spring till autumn with some winter events. Considering the number of regional rain gauges (see Figure 1) and possible deficiencies in data availability, the number of radar-gauge coupled estimates is about 2100 per day.

As mentioned, radar QPE is performed in the composite domain of Figure 5, but it is verified only in the Abruzzo region where the overlap of the nominal maximum unambiguous range of the 3 radars is optimal. The mosaic radar rainfall hourly rain rates $R_{WRNet}(x,y,t)$, given in (6) and obtained using the retrieved rain rates every 10 min, are summed to derive the time-averaged rain rate $R_{WRNet}\Delta t$, i.e.,:

$$R_{WRNet\Delta t}(x,y,t) = \frac{1}{\Delta t}\sum_{i=1}^{N} R_{WRNet}(x,y,t_i)\,\Delta t_i = \frac{1}{N}\sum_{i=1}^{N} R_{WRNet}(x,y,t_i) \tag{7}$$

where N = $\Delta t / \Delta t_i$ are the discrete instants $t_i$ with a period $\Delta t_i$, expressed in minutes, within the integration time $\Delta t$. In our case, N is equal to 6, since $\Delta t_i = 10$ min is constant and $\Delta t = 60$ min (1 h), where in the sum we include the first sample and exclude the last one since radar volumes are labeled with the initial timestamp of each scan. The radar-based hourly rain rates are then compared with the corresponding hourly rain-gauge rain rates, derived in a similar way from the 10-min samples.

The intercomparison is carried out by matching each rain gauge with the corresponding radar mosaic pixel using the nearest neighbor criterion (i.e., each rain gauge measure-

ment is compared with the closest value estimated by the radar around 500 m from the rain gauge itself). Since the minimum rainfall measurement of the rain gauge is 0.2 mm, only rainfall measurements exceeding this value are used for evaluation. The available WR-RG couples are, on average, around 2100 per day and 32,750 in total for all the selected cases.

The results are presented in the following two paragraphs for three selected case studies within those in Table 4 (Section 4.1) and for the whole dataset in terms of conventional statistical error indices (Section 4.2), respectively. Defining the statistical error $\varepsilon_R$ as the difference between the hourly precipitation estimates from weather radar composite $R_{WRNet}$ and the ones $R_{RG}$ measured from rain gauges and indicating with angle brackets the space-time average (for all rain gauges sites and all-time samples), we can compute:

Mean error or error bias (optimum value = 0)     $Bias = <\varepsilon_R> = <(R_{WRNet} - R_{RG})>$

Error standard deviation STD (optimum value = 0)     $STD = \sigma\varepsilon = \sqrt{\left\langle (\varepsilon_R - \langle \varepsilon_R \rangle)^2 \right\rangle}$

Absolute mean error (MAE) (optimum value = 0)     $MAE = <|\varepsilon_R|>$

Root mean square error (RMSE) (optimum value = 0):     $RMSE = \sqrt{\left\langle \varepsilon_R^2 \right\rangle}$

Normalized RMSE (optimum value = 0):     $NRMSE = \sqrt{\left\langle \left( \frac{\varepsilon_R}{R_{RG}} \right)^2 \right\rangle}$

Fractional standard error (FSE) (optimum value = 0):     $FSE = RMSE/<R_{RG}>$
Coefficient of correlation (Corr) (optimum value = 1):     $Corr = \sigma_{WRRG}/(\sigma_{WR}\sigma_{RG})$
Mean–field ratio bias (MRB) (optimum value = 1):     $MRB = <R_{WRNet}>/<R_{RG}>$

Where $\sigma_{WRRG}$, $\sigma_{WR}$, and $\sigma_{RG}$ indicate in *Corr*, respectively, the covariance of the radar-based estimates and rain gauge data, the standard deviation of the radar-based estimates $R_{WRNet}$, and the standard deviation of the rain-gauge measurements $R_{RG}$. Note that RMSE is also equal to the square root of the $Bias^2$ plus the error variance $\sigma^2\varepsilon$ or squared standard deviation (STD). Bias, MAE, RMSE, and STD are in mm/h, being hourly precipitation rates, whereas all other indices are adimensional.

## 4.1. Overview of Selected Case Studies

The selection of the case studies is useful to illustrate and discuss precipitating events that occurred inland and along the coast of the Abruzzo region in different seasons with different weather regimes to highlight the advantages of the composite compared to the observations of a single radar.

The list of the selected events is the following.

1.  *Event on 8 June 2018*. The first case corresponds to a strong inland atmospheric instability that developed into several convective precipitation phenomena. A minimum depression, located on the western Mediterranean, favored the transport of unstable currents over central and northern Italy, resulting in a phase of bad weather, rapidly evolving. It was characterized by precipitation with a predominantly rain shower or thunderstorm character, of strong intensity, with frequent electrical activity, local hail storms, and strong wind gusts.

2.  *Event on 4 February 2020*. A second case examined is characterized by convective phenomena located mainly along the Adriatic coast. The passage of a cold core from northern Europe was responsible for a general and significant drop in temperature and for a reinforcement of the ventilation at all altitudes. The flow of cold air over the Adriatic Sea has also led to the formation of consistent cloud cover associated with showers and locally strong thunderstorms.

3.  *Event on 7 October 2020*. The third case is a widespread event during which the transit of a cloudy system of Atlantic origin through the central Italian regions facilitated showers on the Apennine areas, scattered rains, and the possibility of thunderstorms in the hilly and coastal areas.

Regarding the selected case studies, Figures 6a, 7a and 8a show the mosaicked cumulative surface rainfall total (SRT) with the average technique (2b in Table 3), over a 24-h time interval, of the instantaneous radar-based estimates, whereas Figures 6b, 7b and 8b

depict the cumulative results derived from the rain-gauge network, spatially interpolated using the Kriging method and fitted to the intercomparison domain. The previous figures highlight the different features of each precipitation event with a larger dynamical range of value on 8 June 2018, widespread low-to-moderate rainfall on 4 February 2020, and an intermediate regime with embedded convection on 7 October 2020. Similarities and differences between the radar-based and rain-gauge derived rainfall maps are also noted, such as the higher spatial detail in the radar-based retrievals due to the better WR range sampling with respect to the low-resolution spatially-interpolated rain-gauge maps. Especially for the event on 8 June 2018, in Figure 6a in its northern part, the transition between MM and TO rainfall estimates is sharper mainly due to the X-band rain-rate underestimation in those regions caused by partial beam blockage and ineffective path attenuation mitigation.

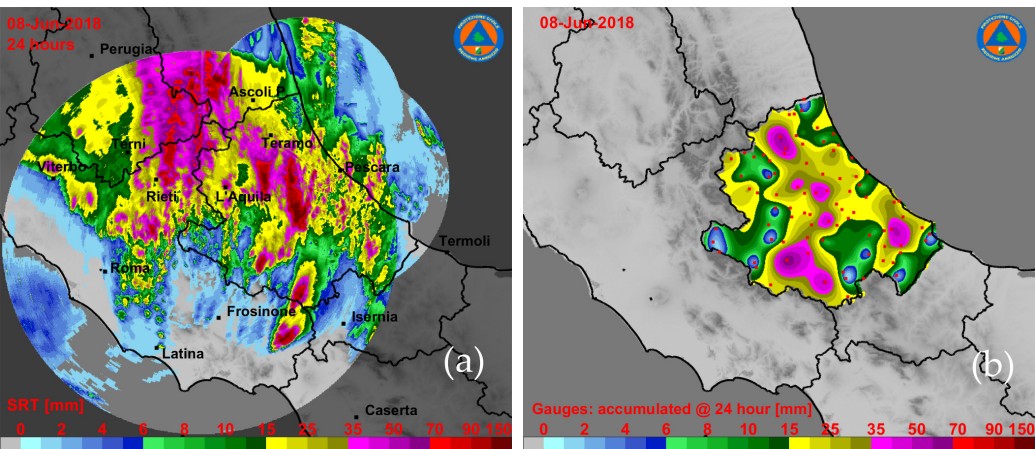

**Figure 6.** 24-h rainfall accumulated during the event occurring during 8 June, 2018: (**a**) radar composite domain and a (**b**) rain-gauge network within the Abruzzo region (see Figure 1).

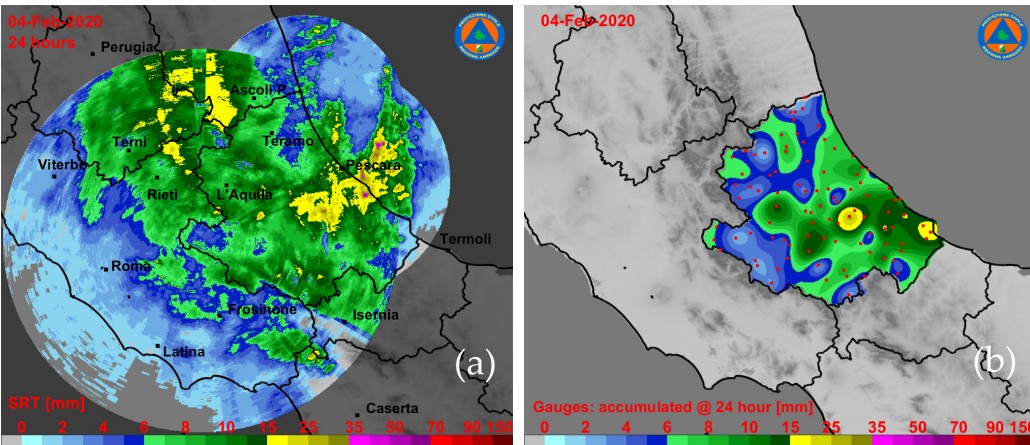

**Figure 7.** 24-h rainfall accumulated during the event occurring during 4 February, 2020: (**a**) radar composite domain and the (**b**) rain-gauge network within the Abruzzo region (see Figure 1).

As mentioned, the 2D products, generated by 3D observations of every single radar in the RAMP processing chain, are remapped in Cartesian coordinates by means of the CRAMS algorithm for the composite generation. As an example, we show CRAMS mosaicked products for the hailstorm, which occurred on 10 July 2019, along the Abruzzo coastline, listed in Table 4. Details on the cyclogenesis of this unusual hail-bearing storm, with supercell characteristics, are provided and discussed in [39]. Figure 9 shows for this event some composite products described in Table 2: vertically maximum intensity (a), rainfall rate (b), vertically integrated liquid (c) and probability of hail (d). The POH

composite map evidences the presence of hail near the Adriatic coast, confirmed by ground observations, as well as high values of vertically integrated liquid [39].

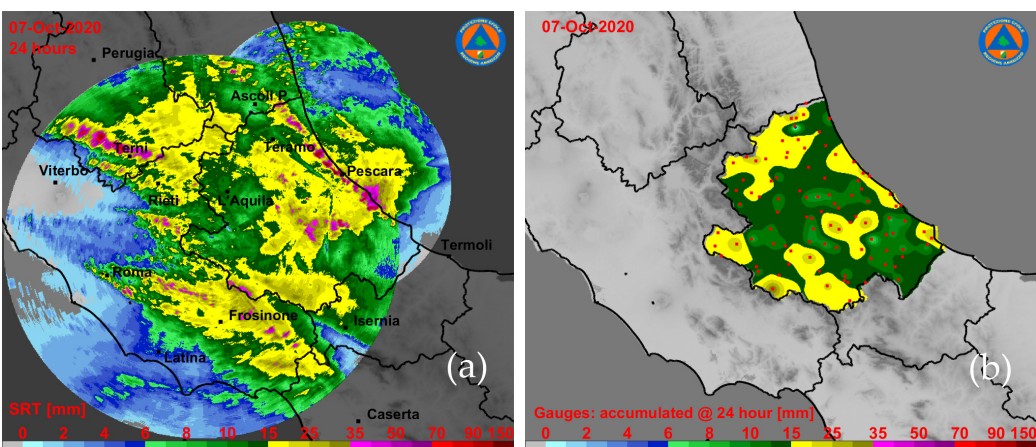

**Figure 8.** 24-h rainfall accumulated during the event occurring on 7 October, 2020: (**a**) radar composite domain and the (**b**) rain-gauge network within the Abruzzo region (see Figure 1).

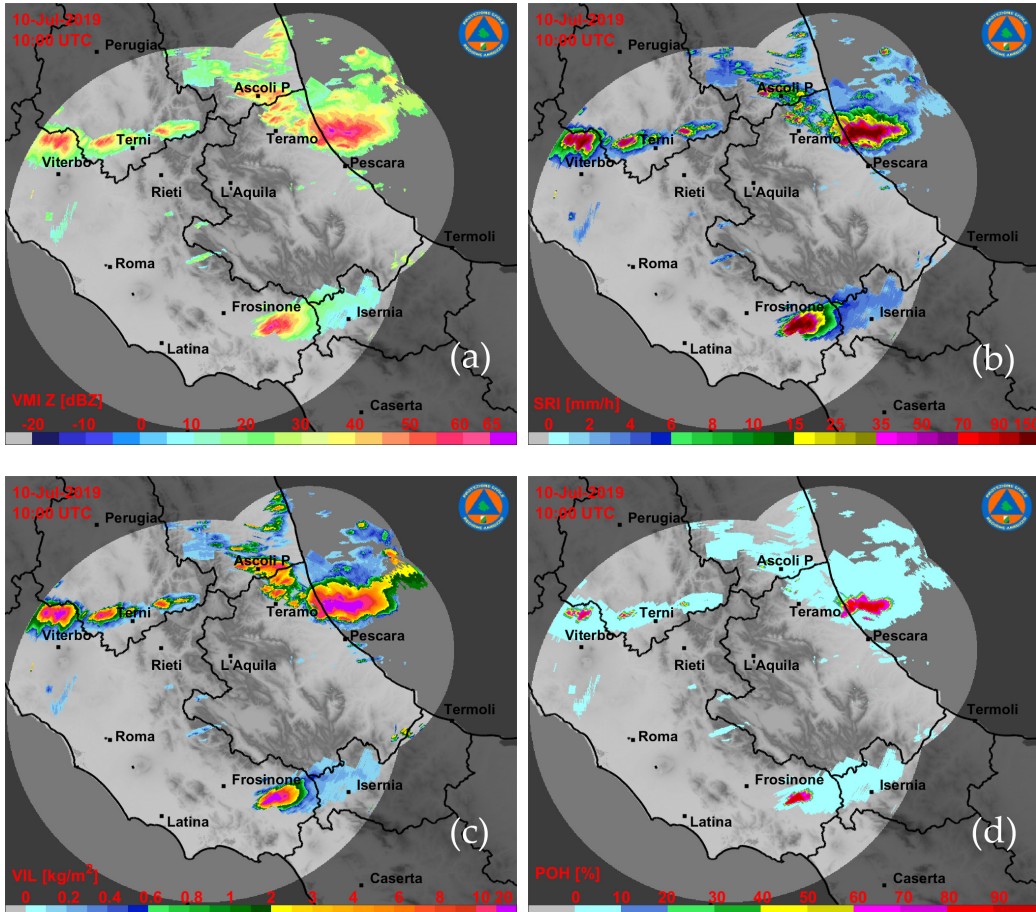

**Figure 9.** Composite radar products generated by CRAMS for the event occurring on 10 July, 2019: (**a**) VMI, (**b**) SRI, (**c**) VIL, (**d**) POH (see Table 2).

*4.2. Analysis of Selected Case Studies*

Considering an hourly time period, the WR-based rainfall estimates can be compared to rain-gauge acquisitions by showing:

-   the rainfall temporal accumulation $C_{WRNet}(t)$ (in mm), also called surface rainfall total (SRT), of hourly radar-based estimates and the corresponding ones $C_{RG}(t)$ of rain-gauge hourly measurements for each mosaicking type of Table 4. In formulas we have:

$$C_{WRNet}(x,y,t) = \int_0^t R_{WRNet}(x,y,t')dt', \quad C_{RG}(x,y,t) = \int_0^t R_{RG}(x,y,t')dt' \quad (8)$$

where $t$ goes from the beginning till the end of the precipitation event.

-   the correlation diagram between hourly rain rate estimates $R_{WRNet}(x,y,t)$ (in mm/h) from weather radar mosaic and hourly rain rate measurements $R_{RG}(x,y,t)$ (in mm/h) from rain gauges.

Figures 10a, 11a and 12a show the rain-gauge cumulative sum $C_{RG}(t)$ (red curve) and the Mt. Midia radar one $C_{WR}(t)$ (green curve) intercomparing all the mosaicking methods, listed in Table 3, with the MM C-band single-radar rainfall estimates (see Equation (A5)). In Figures 10b, 11b and 12b, the hourly rain-rate scatterplot between $R_{RG}$ (from the rain gauge) and $R_{WRNet}$ (from the weather radar network) are plotted using the average method 2b in Table 3 and indicating the normalized sample density through a false-color bar. Note that a threshold (0.2 mm) is set to discriminate the accumulated above the minimum amount detectable by the rain gauges.

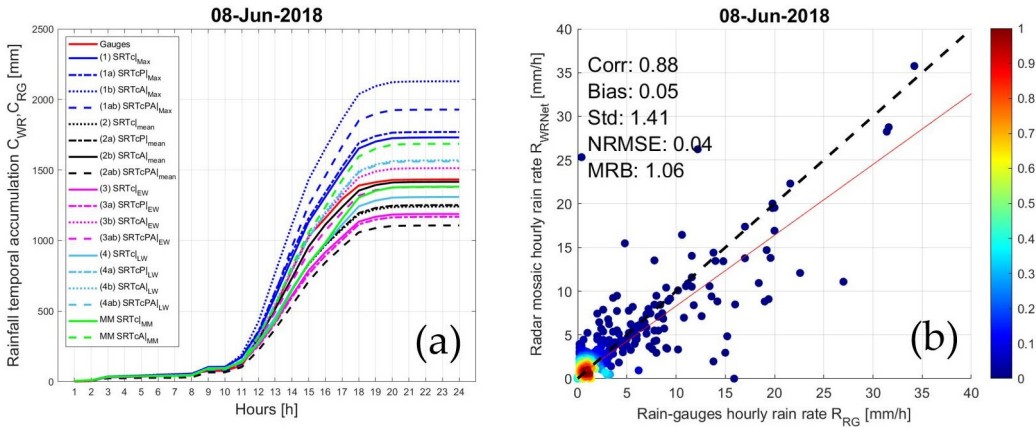

**Figure 10.** (**a**) Time series of rainfall temporal accumulation (see Equation (7)), based on rain-gauge data ($C_{RG}$) and weather radar mosaic data ($C_{WRNet}$) with different mosaicking techniques, for the event occurring on 8 June, 2018. For comparison, the rainfall temporal accumulation from MM single-radar (with and without anisotropic correction) is also shown. See Table 3 for the label legend meaning. (**b**) Scatterplot of radar mosaic hourly rain rate estimates $R_{WRNet}$ (see Equation (6)) and rain-gauge hourly rain rate $R_{RG}$ during the same event. The hourly rain rate $R_{WRNet}$ (see Equation (5)) from the weather radar mosaic is obtained by means of the average method (label 2b in Table 3).

The maximum-value approach (ID = 1 in Table 3 with all its variants, blue lines) is a conventional choice for a radar composite. However, unreliable data are easily passed through, thus introducing a larger uncertainty. There is also a tendency to overestimate precipitation estimates in the presence of convective events. The weighted-value methods with the distance (in Table 3 ID = 3, light blue lines, and ID = 4, magenta lines with all their variants) tend to smooth the rainfall peaks in convective cells and produce a general underestimate of rain-gauge estimates.

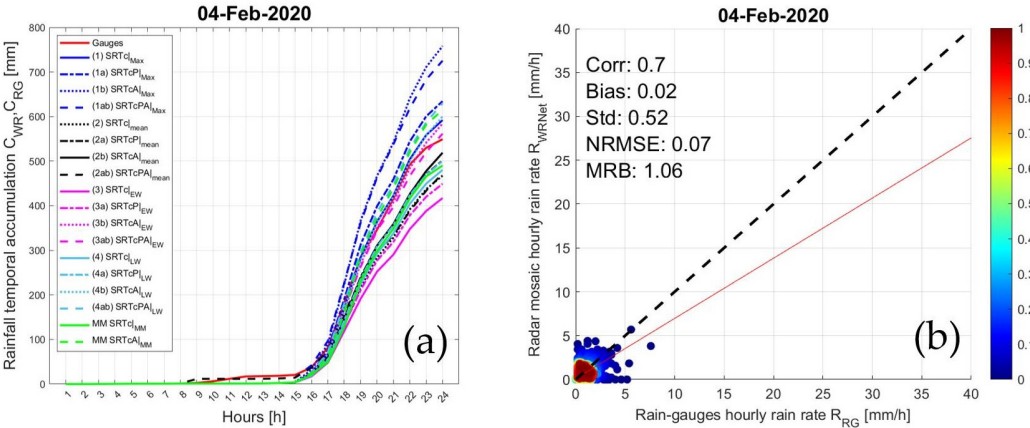

**Figure 11.** As in Figure 10, but for the event occurring on 4 February 2020.

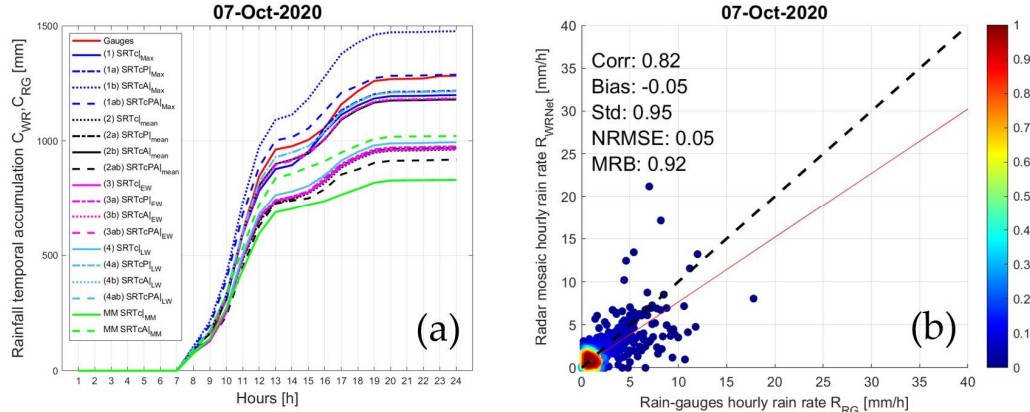

**Figure 12.** As in Figure 10, but for the event occurring on 7 October 2020.

From these figures, it emerges that the use of the mosaic strategy by averaging the radar data after the adjustment factor application is particularly effective. The average-value method (ID = 2 in Table 3, black lines) has the advantage of compensating possible data discrepancy, and at the same time, it tends to equalize the values, especially in the presence of convective cells.

*4.3. Overall Statistical Error Analysis*

After comparing the radar estimates on individual rain gauges for some case studies, the whole event data set of Table 4 is considered to corroborate the previous considerations on a more robust statistical basis. The rainy events analyzed have been collected between January 2018 and December 2020, and the total number of samples is about 32,750. Note that the intercomparison is carried out only within the Abruzzo rain gauge network area only (see Figures 1 and 5) so that C-band MM western observations over the Lazio region and anomalous effects of X-band TO weather radar estimates at longer ranges (as noted in Figure 8) are avoided.

An indication of the deviation of radar estimates from the rain-gauge value is deduced from the above Figure 13, showing the hourly rain rate for the whole data set of Table 4 with the color indicating the normalized sample density. Figure 13a shows the radar–gauges scatter plot of the Mt. Midia single radar, while Figure 13b shows the scatter plot of the composite made with the average-value approach (ID = 2b in Table 3), which proves to be quite accurate in this study.

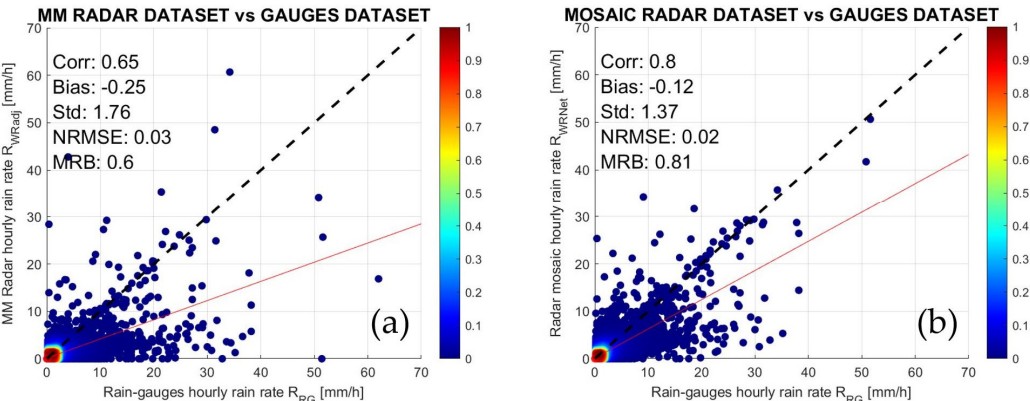

**Figure 13.** (**a**) Scatterplot of M. Midia radar and rain-gauge hourly rain rates during all events listed in Table 4; (**b**) scatterplot of the hourly rain rates between radar mosaic estimates, $R_{WRNet}$, obtained using the average method (2b in Table 3), and rain-gauge estimates, $R_{RG}$.

The most effective way to evaluate the overall performances of the various mosaicking methods using the overall dataset is to compute the statistical error indices, defined previously in this section. Table 5 shows these Corr, Bias, STD, MAE, NRMSE, FSE, and MRB indices for all methods in Table 3. Colors in the table indicate the worst results (red) and the best results (green) for each considered error index (column). The table confirms that approach 2b is the best-performing one, especially for the error uncertainty (i.e., Corr, STD, NRMSE, FSE), with slightly worse results for the mean-error related indices (i.e., Bias, MAE, MRB). Bias, MAE, and STD in mm/h, whereas all other indices are adimensional.

**Table 5.** Overall error indices of the merging methods, given in Table 3, tested in the composite domain using the whole dataset listed in Table 4 (note that MMa stands for MM anisotropic technique). The green and red numbers indicate the best and worst technique for each statistical index (column), whereas the grey row indicates the overall best performing technique (2b, AvgAni).

| ID | Corr Adim | Bias (mm/h) | STD (mm/h) | MAE (mm/h) | Nrmse Adim | FSE Adim | MRB Adim |
|---|---|---|---|---|---|---|---|
| 1 | 0.754 | −0.117 | 1.546 | 0.377 | 0.025 | 2.409 | 0.819 |
| 1a | 0.727 | **−0.093** | 1.659 | 0.390 | 0.027 | 2.581 | 0.856 |
| 1b | 0.747 | 0.030 | 1.732 | 0.417 | 0.028 | 2.685 | 1.047 |
| 1ab | 0.713 | −0.0125 | **1.855** | 0.412 | **0.030** | **2.978** | **0.980** |
| 2 | 0.750 | −0.238 | 1.570 | 0.375 | 0.026 | 2.470 | 0.630 |
| 2a | 0.744 | −0.226 | 1.577 | 0.376 | 0.026 | 2.477 | 0.648 |
| 2b | **0.802** | −0.122 | **1.370** | **0.356** | **0.022** | **2.163** | 0.808 |
| 2ab | 0.677 | −0.251 | 1.680 | 0.418 | 0.027 | 2.749 | 0.594 |
| 3 | 0.740 | −0.253 | 1.587 | 0.373 | 0.026 | 2.509 | 0.605 |
| 3a | 0.733 | −0.241 | 1.594 | 0.373 | 0.026 | 2.515 | 0.624 |
| 3b | 0.760 | −0.155 | 1.511 | **0.356** | 0.025 | 2.361 | 0.759 |
| 3ab | 0.747 | −0.140 | 1.551 | 0.362 | 0.025 | 2.419 | 0.782 |
| 4 | 0.744 | −0.227 | 1.572 | 0.375 | 0.026 | 2.473 | 0.647 |
| 4a | 0.743 | −0.219 | 1.571 | 0.374 | 0.026 | 2.467 | 0.659 |
| 4b | 0.768 | −0.116 | 1.492 | 0.365 | 0.024 | 2.322 | 0.820 |
| 4ab | 0.765 | −0.108 | 1.501 | 0.367 | 0.024 | 2.333 | 0.833 |
| MM | **0.650** | **−0.255** | **1.764** | 0.442 | 0.029 | 2.784 | **0.602** |
| MMa | 0.663 | −0.146 | 1.763 | **0.449** | 0.029 | 2.761 | 0.773 |

Figure 14 shows the same error indices of Table 5, graphed by means of a bar diagram. They are grouped by the mosaicking method (ID) from 1 to 4 and all compared with the performance of the single MM radar (green bar). This representation is very effective to visually comparing the statistical error results. This figure suggests that the corrected radar QPE, using an adjustment factor, has improved performance but only if mitigated through spatial averaging. Method 2b (i.e., assign the mean value among the available measurements after a single-radar spatial anisotropic correction) is among the mosaicking methods with the highest score, as already pointed out. The use of the maximum-value method in the realization of the radar composite tends to emphasize the extreme phenomena leading to an overestimated QPE in the case of convective events.

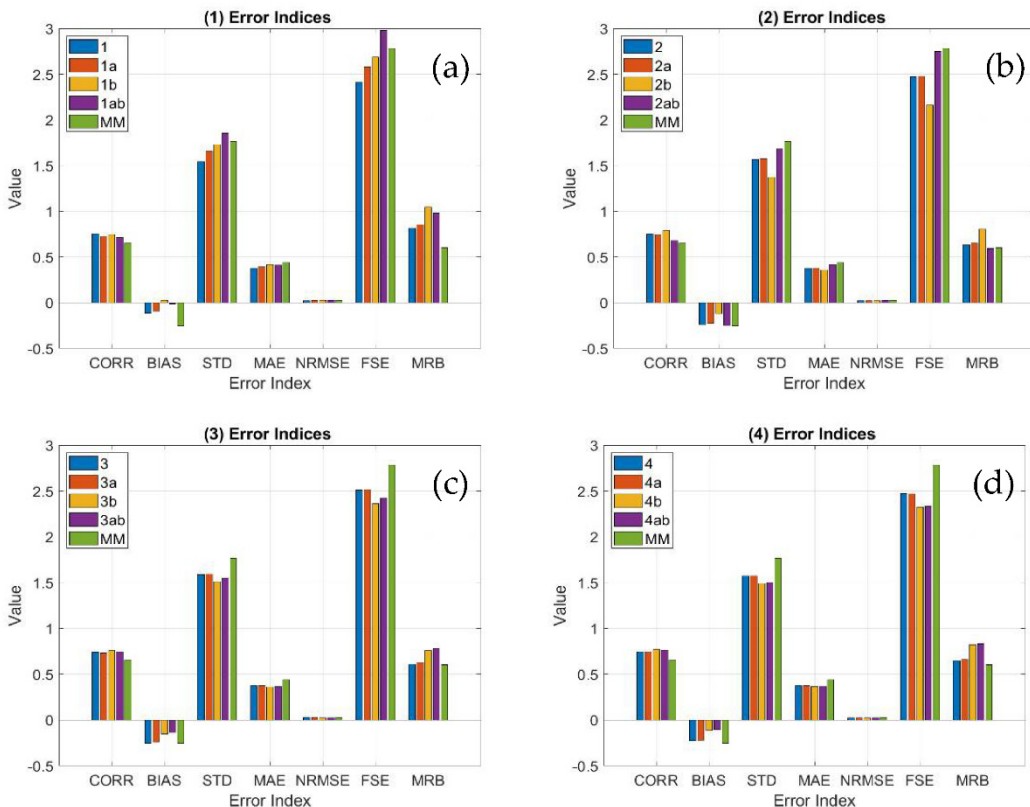

**Figure 14.** Overall error indices of both the tested merging methods and the single radar. (**a**) ID 1-1ab, (**b**) ID 2-2ab, (**c**) ID 3-3ab, (**d**) ID 4-4ab (see Table 3), tested in the composite domain using the whole dataset listed in Table 4.

## 5. Conclusions

A weather-radar network composite algorithm, called CRAMS, has been presented and its rainfall estimates have been tested using two years of rain-gauge data over the Abruzzo region in Central Italy. CRAMS is able to ingest and process data from different radar systems to generate various composite products. It is currently operational in a real-time mode for the Abruzzo region radar network, a heterogeneous network constituted by systems at the C and X bands with and without polarimetric capabilities operating in the complex topography of the Italian Apennines. CRAMS routinely collects every 10 min the available radar data and elaborates composite products displaying them on a web interface.

The weather radar composite has the potential to significantly improve the accuracy and applicability of radar rainfall estimates compared to single-radar QPEs and rain-gauge networks alone. CRAMS can enhance the regional-scale precipitation monitoring in an operational mode to exploit a heterogeneous radar network topology, simultaneously satisfying the needs of multiple users. Its features are: (i) capability of ingesting the 2D radar products of single radars, processed by the RAMP algorithm suite; (ii) applicability

to weather radar networks with different specifications (frequency, beam-width, maximum range, polarization); (iii) suitability for real-time applications thanks to its computational efficiency, widely used in the radar community for its expressive and easy-to-read syntax.

The considered mosaicking methods can take into account spatial radar-gauges adjustment, as well as different spatial combination approaches. A data set of 16 precipitation events during the years 2018–2020 in the central Apennines has been collected (with a total number of 32,750 samples) to show the potential and limitations of the considered operational mosaicking approaches, using a geospatially interpolated dense network of regional rain gauges as a benchmark. Results show that the radar-network pattern mosaicking, based on the anisotropic radar–gauge adjustment and the spatial averaging of composite data, is better than the conventional maximum-value merging approach. The overall analysis confirms that heterogeneous weather radar mosaicking can overcome the issues of single-frequency fixed radars in mountainous areas, guaranteeing a better spatial coverage and a more uniform rainfall estimation accuracy over the area of interest.

This study suggests that storm type, which is highly related to seasons, is one of the main factors affecting the performance of merging methods for operational purposes.

As an example, in the cold season, a possible restraint is the effect of the bright band, which is not compensated for at the single-radar level in this study. Moreover, further and more robust analysis in terms of statistical significance will be needed to corroborate our findings.

Future work should be aimed at evaluating the possible improvement due to mosaicking 3D volume products, instead of 2D ones, extending this approach to nowcasting applications. The data-quality assessment of single-radar estimates remains the most critical issue in a regional composite, even though the combination of multiple measurements can be exploited to better assess the WR mosaic quality itself.

**Author Contributions:** Conceptualization, S.B., E.P., S.D.F., R.L., F.L.R. and F.S.M.; methodology, S.B., E.P. and F.S.M.; software, S.B. and S.D.F.; validation, S.B.; formal analysis, E.P. and F.S.M.; investigation, E.P. and S.D.F.; resources, S.B.; data curation, S.B.; writing—original draft preparation, S.B.; writing—review and editing, F.S.M. and E.P.; visualization, S.B.; supervision, F.S.M.; project administration, F.S.M.; funding acquisition, F.S.M. All authors have read and agreed to the published version of the manuscript.

**Funding:** This research was funded by the Civil Protection Functional Centre of Abruzzo Region (CFA) under the agreement, signed on 28 November 2019, between CETEMPS and CFA.

**Institutional Review Board Statement:** Not applicable.

**Informed Consent Statement:** Not applicable.

**Data Availability Statement:** Radar data are provided by the Civil Protection Functional Centre of Abruzzo Region (CFA). Rain-gauge data are provided by DEWETRA data portal (http://www.mydewetra.org/: 5 November 2021), and the platform is accessible upon request to the Italian Civil Protection Department (DPC).

**Acknowledgments:** The authors are thankful to all colleagues of the Civil Protection Functional Centre of Abruzzo Region (CFA) and to HIMET Srl (L'Aquila, Italy) for providing data processing infrastructures.

**Conflicts of Interest:** The authors declare no conflict of interest.

## Appendix A. Radar Advanced Multiband Processing (RAMP) Main Modules

The 3D volume scans of each radar system are processed by means of the radar advanced multiband processing (RAMP) processing chain. All RAMP algorithms are organized in a series of sub-packages or steps, each characterized by its own specific function and operating in native radar (spherical) coordinates. The steps are briefly described in the following sections.

*Appendix A.1. Pre-Processing Correction*

For each radar, a pre-processing correction of volumetric data is applied. Ground-clutter, specks, wireless local area network (WLAN) interference, and biological and other non-meteorological echoes are removed from the horizontally polarized reflectivity $Z_{hh}$ (or in dBZ expressed as $Z_{HH}$) field by exploiting the textural spatial correlation of meteorological targets with respect to artifacts [28]. Uncertain data are flagged but not removed. For dual-polarization systems, in addition, a median smoothing filter and bias correction are applied to the differential reflectivity $Z_{dr}$ (or in dB expressed as $Z_{DR}$), whereas compensation is applied to the correlation coefficient ($\rho_{HV}$). The differential phase $\Phi_{dp}$ (in °) is processed and then reconstructed by a multistage smoothing filter for estimating the differential specific phase shift $K_{dp}$ (in °/km) [40].

*Appendix A.2. Partial Beam Blockage Correction*

Due to the complex topography of the Abruzzo region, characterized by high mountains, all radar systems suffer from beam blockage along several directions, especially at lower elevation scans. The second step of the single-radar processing chain is the partial beam blockage (PBB) correction. When a radar beam intercepts an obstacle, two situations are possible: (1) only part of the beam cross-section illuminates the intercepted topography (partial blockage); (2) the radar beam is completely blocked (total blockage). Occlusion of the beam less than 10% is considered negligible and is not corrected, whereas an occlusion exceeding 60% is rejected. If radar bins are partially shielded with a blockage between 10% and 60%, the radar reflectivity factor measurements are modified by adding 1–4 dB depending on the degree of occultation [41].

*Appendix A.3. Path Attenuation Correction*

The horizontal radar reflectivity $Z_{HH}$ is proportional to the received radiation backscattered by the raindrop size distribution (DSD) within each radar volume bin, whereas $Z_{DR}$ is the difference between $Z_{HH}$ and $Z_{VV}$. The polarimetric attenuation correction algorithms are applied to $Z_{HH}$ and $Z_{DR}$ (corrected by the first step and PBB correction modules), using the reconstructed differential phase shift $\Phi_{dp}$ [42]. If PIA indicates the 2-way path integrated attenuation (in dB), the range differences of the reconstructed phase between two cells are connected to the total path attenuation increments along the path through the coefficients $\alpha$ and $\beta$ [30]:

$$Z_{HHc} = Z_{HH} + PIA = Z_{HH} + \alpha \left[ \Phi_{dp}(r_N) - \Phi_{dp}(r_0) \right] \tag{A1}$$

$$Z_{DRc} = Z_{DR} + PIA = Z_{DR} + \beta \left[ \Phi_{dp}(r_N) - \Phi_{dp}(r_0) \right] \tag{A2}$$

where $\alpha$ = 0.28, $\beta$ = 0.04 for the X band [43], $\alpha$ = 0.08, $\beta$ = 0.03 for the C band [44], and $r_N$ and $r_0$ are, respectively, the farthest cell and the initial cell along the radar beam path. In (1) and (2), the terms $Z_{HHc}$ and $Z_{DRc}$ indicate the corrected quantities, whereas *PIA* is the two-way integrated path attenuation that undergoes the radar signal in the presence of precipitation between the cells at range $r_0$ and $r_N$. For single-polarization systems, *PIA* is estimated using an iterative procedure using only the measurement of horizontal reflectivity $Z_{HH}$ and an arbitrary power-law between $Z_{HH}$ and *PIA* [29]. However, this scheme is notoriously unstable; thus, in this case, a threshold value is set on *PIA* to limit the correction itself.

It is worth mentioning that nowadays X-band radars represent a cost-effective solution where the domain of interest is limited in size to a county because these radars are much cheaper (in terms of direct costs, infrastructure, and maintenance) than traditional C-band or S-band ones. On the other hand, at the X band, path attenuation due to rainfall can be quite significant so that, in case of heavy rain, we can have even a total signal loss. This problem can be only addressed by providing additional radar information from a separate source that is not experiencing a total signal loss. This is why radar networks can become useful especially at the X band.

*Appendix A.4. Total Quality Information*

The quantitative estimation of error magnitude is necessary not only in order to gain general knowledge about data uncertainty but also to apply quality information in further data processing, e.g., in the generation of standard or user-related specific products [29,45]. One of the most common approaches in the characterization of weather radar data quality is to employ a quality index $q_i$, defined as a unitless quantity providing data reliability on a digital scale.

The idea of the radar data quality scheme is based on the selection of quality factors, determination of their quality indices, and the computation of one total quality index. Assuming that each radar system is well maintained, the quality analysis is focused on the error sources resumed in Table A1. For each error i, a quality index $q_i$ is estimated in each radar pixel through appropriate tests, giving as output a unitless quantity from 0 (bad quality) to 1 (excellent quality).

The total quality index (TQI) in each radar pixel can be retrieved by combining all the individual quality indicators listed in Table A1:

$$TQI = q_{SYS} \; q_{NME} \; q_{PBB} \; q_{RAN} \; q_{ATT} \; q_{VPR} \tag{A3}$$

where a multiplicative combination rule is here used as in [45].

**Table A1.** Group of quality indices $q_i$ and related sources of error.

| Quality Index | Source of Error | Note |
|---|---|---|
| $q_{SYS}$ | Radar system technical parameters | It is static within the whole radar range as well as in time and takes into account several factors as in [29]. |
| $q_{NME}$ | Non-meteorological echo | Pixels affected by non-meteorological echoes are removed; for the uncertain pixel a value of 0.5 is applied, and the other data are set to 1. |
| $q_{PBB}$ | Partial beam blocking | It is computed from the corrected data taking into account the PBB value as in [28]. |
| $q_{RAN}$ | Long-range measurement | This quality factor decreases with increasing the measurement distance from the radar; it is computed as in [45]. |
| $q_{ATT}$ | Rain path attenuation | It is computed from the corrected data taking into account the *PIA* value as in [28]. |
| $q_{VPR}$ | Inhomogeneous vertical profile of reflectivity | The compensation of this effect is not performed in RAMP; the associated quality index is estimated as in [46]. |

*Appendix A.5. Rainfall Rate Estimation*

After performing the previous corrections, the near-surface rainfall rate $R$ (in mm/h), also called surface rainfall intensity (SRI), is computed from the vertical maximum intensity (VMI) or from the sweep at first elevation or the lowest detectable bin (LDB) by means of the following estimators. In particular, for C- and X-band polarimetric radar we have:

$$R(Z_{HHc}, Z_{DR}) = \alpha \; 10^{(0.1 \cdot \beta \cdot Z_{HHc})} \; 10^{(0.1 \cdot \gamma \cdot Z_{DR})} \tag{A4}$$

with $\alpha = 0.00899$, $\beta = 0.927$, and $\gamma = -5.05$ derived from disdrometer data [47], whereas for the single-polarization radar we have:

$$R(Z_{HHc}) = \left( \frac{1}{\alpha} 10^{(0.1 \cdot Z_{HHc})} \right)^{\frac{1}{\beta}} \tag{A5}$$

with $\alpha = 443.5$ and $\beta = 1.2987$ [47] for X-band single-polarization radar and with $\alpha$ and $\beta$ values as in [8] for C-band single-polarization radar. In (A4) and (A5) $Z_{HHc}$, $Z_{DR}$, and $R$ are in dBZ, dB, and mm/h, respectively. CRAMS also foresees parametric relationships, as in (A4), including $K_{dp}$ where its estimate is not too noisy [40]. Recently, some studies have focused on developing a polarimetric QPE based on specific attenuation $A$ (in dB/km) [48,49].

However, the estimator $R(A)$ must be very carefully used in the situations of very light and sporadic rain wherein the attenuation signal is too weak and in widespread light stratiform rain [50]. The potential of using $A$ might be limited depending on the approach to obtain $\Phi_{DP}$ [30,51]). Although the radial interval to calculate A can be freely selected, $\Delta\Phi_{dp}$ could be inaccurate at short path intervals and/or be contaminated by backscatter differential phase $\delta hv$, as a result of Mie scattering and random fluctuations. For these reasons, CRAMS is not ingesting algorithms based on $R(A)$.

From the estimated $R$, the accumulated rain, named SRT (surface rainfall total) can be computed, at a given generic time interval, by two steps as described below. First, the accumulated hourly precipitation, at time $h$, is computed by the following relationship:

$$R_{h,1} = \frac{1}{M} \sum_{i=1}^{M} R_i \tag{A6}$$

where $M$ represents the available rainfall rate $R$ in the hourly time interval (6 for Abruzzo radars). Then, the accumulated precipitation between hours $h$-$N$ and $h$, or the accumulated at $N$ hours from the current time $h$ is computed by:

$$C_{h,N} = \sum_{i=0}^{N-1} R_{h-i,1} \tag{A7}$$

*Appendix A.6. Vertically Integrated Liquid Estimation*

An excellent indicator of severe storm activity is the VIL (vertically integrated liquid) product, especially with regard to the rainfall potential of a storm. In general, it can be considered a means of locating the most active and severe storms in a region; in stratiform situations VIL rarely exceeds a value of 10 kg/m$^2$, in thunderstorms, however, VIL is usually (much) higher.

VIL is calculated by vertically integrating reflectivity values from the top of a thunderstorm to the ground and converting reflectivity data into an equivalent liquid water content value [52]. The three-dimensional radar data is converted to a map of the amount of liquid water present in a vertical column above a certain position. The general equation for VIL (expressed as $L_V$ in kg/m$^2$) is:

$$L_V = \sum_i a \left[ \frac{(Z_i + Z_{i+1})}{2} \right]^b \Delta h \tag{A8}$$

where $Z_i$ and $Z_{i+1}$ are two adjacent radar reflectivity bin values (expressed in mm$^6$ m$^{-3}$) at the same horizontal coordinate, and $\Delta h$ is the vertical distance between the *i-th* and the *i + 1-th* bin (expressed in meters). The pair $a$ and $b$ of coefficients are set as a function of the frequency band (e.g., [30,39]).

When calculating VIL, has to be taken into consideration that its values in storms located too close to the radar will be underestimated, because the radar is not capable of scanning high enough to reach the upper portions of the storms.

*Appendix A.7. Probability of Hail*

Hail events are typically related to crop losses, building and car damage, and casualties. Nowadays, the simplest and direct way to distinguish between hail and rain is by using radar reflectivity. This is an alternative to the classification that requires good-quality polarimetric parameters. Several methods that use radar reflectivity and other ancillary data are available from literature and some of them were tested by [53]. Among these, for the needs of the Abruzzo Region, a method that uses the value of VIL density (VLD) has been accomplished [54]. VLD is simply the VIL divided by the EchoTOP and multiplied by 1000 in order to express the result as g/m$^3$. We remember that the EchoTOP is the height of the highest (in altitude) bin measured by a radar during a volumetric scan.

VLD makes VIL independent of height and then reduces errors due to the fact that VIL alone may not be sufficient to distinguish tall storms with low overall reflectivity (smaller targets, including possible small hail) from short storms with high reflectivity (larger targets including possible large hail).

Thus, VLD is well-adapted for the algorithms of hail detection since thunderstorms with larger VLD values generally produce larger hailstones at the surface; usually, VLD values range from 0 to 5 $\text{g·m}^{-3}$. To find a relationship between POH and VLD several thunderstorms with and without hail have to be examined with a radar system. The probability of hail (POH, in %) rises sharply as VLD increases and their relationship can be expressed, as an example, with a third-order polynomial fitted curve [55].

*Appendix A.8. Convective Storm Detection*

Distinguishing between convective and stratiform is an important indicator of the vertical and horizontal structure of cloud systems producing precipitation. This precipitation-identification product is aimed to provide a separation of the radar echoes into convective and stratiform regions, on the basis of the intensity and sharpness of the peaks of echo intensity [56]. The method for the identification of precipitation type is based on the horizontal structure of the precipitation field. The idea consists of the search for the reflectivity peaks: if they satisfy specific criteria, related to the ratio of the peak and the mean reflectivity of their immediate surroundings, they are classified as convective centers. The next step is classifying the area immediately adjacent as intermediate between stratiform and convective (mix), introduced to indicate those events whose nature is uncertain, while the remaining reflectivity field is categorized as stratiform.

*Appendix A.9. Nowcasting*

The nowcasting radar module is aimed to provide an indication of the temporal evolution of reflectivity or rain field at future instants, in particular for convective cell storms. Due to high temporal variability of heavy rain cells (usually lasting for periods larger than the radar temporal resolution $\Delta T$ but less than 30 min), the nowcasting methods often propagate the information at short instants ahead, $t_0 + n \cdot \Delta T$, with respect to the procedure initialization instant $t_0$.

The nowcasting methodology adopted into CRAMS is based on the spectral pyramidal advection radar estimator (SPARE) algorithm by [57]. The procedures take a temporal sequence of available radar maps and propagate the last available one in the future. The principle is based on the cross-correlation between portions of two consecutive radar maps to compute the displacement vector between them [58]. The segmentation of each available radar field (also regarded as a special case of spatial decomposition) is a fundamental step that allows computing the displacement vectors for each identified portion of the radar maps. This implies that the resulting motion field is composed of several vector components (one for each identified portion of the radar map) that in principle are different from each other. Thus, vortex or multiple system movements can in principle be caught.

The original concept of the SPARE algorithm, in its primordial version, is to perform spatial correlation on filtered radar images in the spectral domain of spatial frequency. The spatial filter used in the SPARE algorithm is able to isolate spatial components in a prescribed range of spatial scales. For this reason, SPARE is said to be pyramidal, since the decomposition of radar maps in spatial components resembles a pyramid. This way to proceed tends to ensure a better estimation of the displacement field. The input variable for the radar nowcasting algorithm is the reflectivity at the two consecutive past instants with respect to procedure initialization.

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
