# Peer review of "Mosaicking Weather Radar Retrievals from an Operational Heterogeneous Network at C and X Band for Precipitation Monitoring in Italian Central Apennines"

_remotesensing, doi:10.3390/rs14020248_

Round 1
Reviewer 1 Report
Manuscript Title: Mosaicking weather radar retrievals from an operational heterogeneous network at C and X band for precipitation monitoring in central Italy Apennines
Summary:
This manuscript describes the development of an algorithm to merge weather radar data from different radars (at different frequency bands, different specs/capabilities), including real-time data synchronization and a quality control assessment. Results from a 2-year test period were verified by comparing radar-derived rainfall rates (QPE) against direct rainfall measurements taken with a network of rain gauges. A variety of processing and evaluation metrics are presented. The discussion presents several data collection cases, were one of the merging methods (2b) appears to have overall best performance compared to other metrics.
The paper reads well, summarizes a long-term effort, and will be important for the operational weather radar community. There are a couple of relatively straightforward
Main Minor Comments:
- Please list the limitations of the presented method in the conclusions. There is no review of the challenges/limitations, only positive outcomes. Readers must be aware of the potential limits of the proposed algorithm, in case they pursue operational implementation.
- There are several very long sentences, expressing multiple ideas. It makes it a bit hard to follow the sentence and retain ideas when it’s too long. I suggest shortening some of them by breaking them into 2 or 3 subsentences, when possible. It will greatly improve readability.
More Minor Comments / Suggestions:
- Line 22: “… mosaicking methods can take into account spatial radar-gauges…”. Seems like a word is missing in this sentence. Maybe you meant “take into account”.
- Line 27: “… anisotropic radar-gauges adjustment”. Should be singular.
- Line 79: The sentence starting in this line is a bit hard to understand. What do you mean? Also, I believe “radar observation” should be plural (i.e., observations). Please revise.
- Line 128: This sentence is too long (82 words!) and expresses a couple of ideas. I suggest breaking it up into 2 or 3 shorter sentences.
- Line 161: I’m not sure if the term “radial resolution” is very common. I suggest using “range resolution”, which is more widely used in the community.
- Line 175: A word seems to be missing at the end of the sentence. Perhaps “…and relatively accurate ”.
- Line 176: This is known, but I suggest providing some references here.
- Line 184: I suggest flipping the order of “temporal resolution and accuracy” to be consistent with the order of the previously mentioned “bucket resolution” and “hourly rainfall accumulation time period”.
- Line 232: “…guaranteeing measures measurements even in…”
- Line 255: How are the input and output data highlighted? I don’t see this in the top of Fig. 2.
- Line 268: I don’t think you need “data” at the end of the sentence, since the subject is the products (i.e., “…carried out both for volumetric 3D and 2D products data.”).
- Table 2: The caption says, “each single-polarization radar”, but the X-band system in Tortoreto is dual polarization and is part of the prototype network presented. Please clarify.
- Line 299: “radar-based SRI or hourly rain rate”. Did you mean “of hourly rain rate”?
- Line 333: “in the next paragraph 3.2.”. I suggest rewording simply as “in the next section.”.
- Line 338: Can you add appropriate references to these statistical methods?
- Line 371: Maybe “kriging” should be capitalized, as in previous instances.
- Line 450: In the caption of Table 3, I suggest changing “(from a to b)” to simply “(a or b)”. I like this table because it provides references and is well organized.
- Line 491: The acronym for QPE has been defined previously, it is not necessary to redefine it (it could lead to confusion, since you are re-defining something). Other acronyms were also defined more than once, I suggest revising them.
Author Response
Dear reviewer,
We truly thank for his fruitful contribution.
Please see the attachment for the replies to your observations.
Kind regards,
Stefano Barbieri

Reviewer 2 Report
Overall a good study on evaluating various mosaicking methods as it applies to the mixed radar network of C and X-bands in complex orography. The study demonstrates demonstrates the value of mosaic products for hydrological applications as opposed to single radar products. Some more details/clarification on the algorithms and mosaicking technique would be useful to include in the descriptions, apart from the references.
Comments:
It would be better to see the radar coverage rings and some measure of distances for those unfamiliar with the region on the first figure, rather than having to wait to figure 5.
Lines 189-203: It is clear that preliminary QC of gauge data via typical quality checks was done, however it is not clear if corrections described in 180-196 were applied in the study.
Lines 296-327. This suggest that for rain-rate conversion you used the maximum reflectivity in the column for the overlapping regions. Is this true for non-overlapping regions as well. Can you also be clear on what a,b was used and also the parameters or the polarimetric relation.
On page 16 , consider reversing the order of the description of the events to match how they are displayed in figures 6-8. That is describe Feb 4 before Oct 7.
For the cases displayed, it seems that the discontinuity in rates(with Mt. Midia) are largest for the Tortoreto radar independent of event type. More so than Cepagatti. Can you provide some explanation or potential reasons, calibrations ?, polarimetric estimates?
Was there not enough gauge data or some other reasons not to include the convective hail case in the analysis?
How does the mosaicking algorithm handle radar bright bands in colder season?
Was the Feb 04 case all rain at the gauge locations? Was there a mix of precipitation types at gauge locations at higher altitudes?
Line 673-676: Mosaicking does show an improvement and scatter is still quite large, so I would not say it proves to be accurate. At the very least it is relatively better than the single radar estimates.
Figure 14 and the overall description: True there is a definite improvement of the mosaic vs single radar, but it seems less evident there is one in the choice of mosaicking methods(2,3,4). Perhaps the improvement may become clearer by stratifying the mosaic methods by event types, for example one method might be better in convection than another. Overall, 2b may have a slight advantage over the other methods, but not a clear one.
Bottom line: Anisotropic radar-gauge adjustments and averaging are better than max-value compositing methods, seems appropriate, and would be good to see your future work on this.
Author Response

(The authors gave the same response as above.)
